# Aesthetic Impact of Orthognathic Surgery vs. Orthodontic Camouflage in Class II Division 1 Patients with Convex Facial Profile: A Follow-Up Using Combined Frontal and Profile Views

**DOI:** 10.3390/jcm14124277

**Published:** 2025-06-16

**Authors:** Simos Psomiadis, Iosif Sifakakis, Ioannis Iatrou, Nikolaos Gkantidis

**Affiliations:** 1Department of Oral and Maxillofacial Surgery, School of Dentistry, National and Kapodistrian University of Athens, GR-11527 Athens, Greece; psomiad@dent.uoa.gr (S.P.); iiatrou@dent.uoa.gr (I.I.); 2Department of Orthodontics, School of Dentistry, National and Kapodistrian University of Athens, GR-11527 Athens, Greece; isifak@dent.uoa.gr; 3Department of Orthodontics and Dentofacial Orthopedics, School of Dental Medicine, University of Bern, CH-3010 Bern, Switzerland

**Keywords:** patient outcome assessment, facial appearance, convex profile, dental overjet, orthodontics, orthognathic surgery

## Abstract

**Background/Objectives**: A previous study evaluating convex facial profiles at rest demonstrated that combined orthodontic and orthognathic surgical treatment is more effective in enhancing facial aesthetics compared to orthodontic camouflage alone. The present follow-up study aimed to reassess these findings by incorporating both profile and frontal facial views in the aesthetic evaluation. **Methods**: This retrospective cohort study sample included 36 consecutively selected patients with convex facial profiles and Class II Division 1 malocclusion. Two groups of 18 non-growing patients with similar characteristics were compared. Group A was treated with orthodontics and orthognathic surgery, whereas Group B was treated with orthodontics exclusively. Pre- and post-treatment profile and frontal facial photographs were simultaneously presented to orthodontists, oral and maxillofacial surgeons, convex profile patients, and laypeople, asking them to assess changes in facial appearance. **Results**: Significant positive changes in facial appearance were perceived for Group A, in contrast to no changes for Group B, with a difference of 17/100 visual analogue scale (VAS) units. The rater groups demonstrated a high degree of consistency (ICC > 0.88). Multivariate analysis revealed significant differences in perceived changes between the two treatment groups (F = 14.63, *p* < 0.001, Pillai’s Trace = 0.36, and partial η^2^ = 0.36), with no significant effects from the rater group (*p* > 0.05). Similar results are evident when only profile photos were rated (*p* > 0.05). **Conclusions**: The combined orthodontic and orthognathic surgery approach effectively enhances facial appearance in convex profile cases, whereas orthodontic treatment alone does not result in significant changes. These findings should be clearly communicated during patient consultations and considered in treatment planning.

## 1. Introduction

Facial appearance refers to the visual characteristics of the face, including its shape, symmetry, skin quality, and overall look, and plays a vital role across numerous aspects of life, affecting not only superficial interactions, but also deeper social, psychological, and economic outcomes [1]. In professional settings, facial appearance has been shown to influence hiring decisions, career advancement, and perceptions of leadership. Socially, facial attractiveness is associated with acceptance, popularity, and relationships, impacting social networking and romantic connections [1,2,3]. From a psychological perspective, the self-perceived facial appearance can profoundly impact self-esteem, confidence, and mental health [4]. Concerning the economic implications of facial appearance, more attractive individuals may earn higher wages than their less attractive counterparts and are more persuasive in marketing and advertising, influencing consumer preferences and decisions, particularly for appearance-relevant products [5]. Lastly, the significance of facial appearance in human life also has ethical and cultural dimensions. Ethical debates arise in the context of cosmetic surgery and the societal pressures to conform to certain aesthetic standards. Cultural differences in what is considered attractive emphasize the diversity of human societies and the subjective nature of beauty [6]. Thus, the impact of facial appearance on human life is multifaceted [1,7].

A convex facial profile, frequently associated with Class II malocclusion, is a prevalent skeletal configuration that is often perceived as less aesthetically pleasing, particularly when characterized by a prognathic maxilla and a retrognathic mandible [8,9]. Consequently, patients often seek treatment for this condition, which involves orthodontic interventions, more invasive orthognathic surgery approaches, or a combination of both [10,11]. Enhancing facial appearance is a primary motivator for seeking treatment and is closely tied to patient satisfaction [12,13,14].

The extent to which orthodontic interventions alone can significantly enhance facial appearance has been questioned, even in growing patients. This skepticism is supported by cephalometric data [15,16] as well as facial perception studies [17,18]. Patients with convex profiles, after their growth ceased, primarily have two treatment alternatives. The first option is orthodontic treatment, which focuses on specific modifications within the dentoalveolar structure and is commonly referred to as camouflage orthodontic treatment. This approach relies on the retraction of protruding maxillary incisors to improve both dental occlusion and facial aesthetics, without addressing the underlying skeletal discrepancy. Beyond enhancing dental aesthetics, camouflage orthodontic treatment also aims to optimize dental function, establish proper occlusion, and support long-term oral health [19]. The second option involves orthognathic surgery, a more invasive intervention that also seeks to enhance facial appearance—a key consideration for many patients. However, the tangible benefits derived from each intervention are not always well-defined, contributing to an ongoing debate in the scientific literature that influences patient decision-making and, consequently, treatment planning [19,20,21].

In a prior investigation that assessed treatment effects on facial profile photos, the perception of facial appearance alterations strongly favored the combined orthodontic and orthognathic approach over exclusive orthodontic treatment [20]. Nevertheless, earlier research on convex profile adolescents who received conventional orthodontic appliances suggested that the observed profile improvements largely diminished when frontal and profile facial images were presented simultaneously to the evaluators [17,18]. Thus, we assessed here the facial outcomes of combined orthodontic and orthognathic intervention compared to orthodontic camouflage treatment through the simultaneous presentation of profile and frontal facial photos to rater groups. We hypothesized that Class II Division 1 convex profile patients would exhibit similarly perceivable changes in facial appearance whether treated with a combination of orthognathic and orthodontic treatment or with orthodontic (camouflage) treatment alone, particularly when both the frontal and profile resting views are evaluated. Understanding perceptions across diverse rater groups on this critical issue will aid in informed decision-making and treatment planning for our patients.

## 2. Materials and Methods

The study protocol was approved by the Research Ethics Committee of the Dental School, National and Kapodistrian University of Athens, Greece prior to study commencement (date of approval: 22 June 2018, protocol number: 361). This retrospective comparative study was reported in accordance with the STROBE guidelines for cohort studies (Appendix A). All evaluated patients provided informed consent, allowing their data to be used for research. No eligible patient refused participation. The methods are similar to those of a previous publication from our team where the evaluators rated pairs of profile facial images [20]. In the present study, originally derived data from simultaneous assessments of profile and frontal facial image configurations were analyzed and compared to previously published data on single profile photo ratings of the same sample [20]. Our goal was to determine whether presenting additional facial aspects—beyond the profile, which is the primary treatment target—reduces or eliminates perceived differences in outcomes [20], as observed in another setting [17,18]. Viewing more than one facial aspect is considered a more realistic representation of real-life perception, as people naturally observe faces from multiple angles during daily interactions. Apart from the latter strategy, both studies referred to the same patient sample and all other methodological aspects were applied similarly for comparability reasons. However, most methodological considerations will be repeated here to allow for proper comprehension of the study by the readers.

### 2.1. Sample

The study sample was sourced from the Postgraduate Clinic of the Department of Orthodontics, Dental School, National and Kapodistrian University of Athens, Greece and was identical to that of the previous publication [20]. The sample was selected consecutively from the most recently treated Class II Division 1 patients with a convex facial profile who met the inclusion criteria. The goal was to form two sex-matched groups of 18 patients each (Groups A and B). The sample size was selected based on empirical data, considering also resource constraints and practical feasibility in terms of available patients and number of required raters [17,18,22]. A post-hoc power analysis was conducted using G*Power (version 3.1.9.6) [23] to determine the required sample size for a MANOVA with two independent variables (2 × 4 design) and five dependent variables. The analysis assumed a medium effect size (*f*^2^_v_ = 0.2) based on Cohen’s guidelines [24], an alpha level of 0.05, and a desired statistical power of 0.80. The results indicate that a total sample size of 32 evaluated patients was required. Therefore, the present sample size of 36 patients was deemed adequate for the primary study outcomes. Group A included non-growing patients with a convex profile who received orthodontic treatment with full-fixed appliances in conjunction with mono- or bi-maxillary orthognathic surgery. Group B included non-growing, convex-profile patients who received solely orthodontic treatment with fixed appliances. The specific plans were tailored per case, according to each patient’s needs and demands, and were not considered in sample selection.

Eligibility criteria required patients to have complete initial and final diagnostic records, including medical and dental histories, details of orthodontic or orthognathic treatment, pre-treatment panoramic and cephalometric radiographs, pre- and post-treatment dental models, as well as adequate-quality intraoral and facial photographs. Evaluated patients needed a Class II Division 1 malocclusion before treatment (molar Class II > half cusp in both sides, overjet 6–12 mm, and no functional shift ≥ 2 mm), a convex skeletal configuration (5° < ANB < 9°), and a convex facial profile (males: 15° < facial contour angle < 25°, females: 17° < facial contour angle < 27°) [25]. Additional criteria included an Frankfort mandibular plane angle (FMA) angle of 17.5–32.5°, treatment duration of 1–5 years, no history of aesthetic facial surgery, White European ancestry, no craniofacial anomalies or syndromes, no marked facial asymmetries assessed independently by two authors through visual examination, ceased skeletal growth (CS5-CS6, age over 15 years), a full dental arch without considering the third molars, completed treatment without discontinuation, and no use of fixed mandibular advancement devices [20].

During sample selection, only the initial diagnostic records were utilized, while the final records were reviewed solely to confirm their availability. From each patient’s diagnostic data file, the initial and final lateral and frontal facial photographs were used for the assessment of the perceived changes in facial appearance by the raters. These were captured with the Frankfurt horizontal plane parallel to the floor, the teeth lightly occluded in maximum intercuspation, and the lips in a relaxed position.

### 2.2. Facial Photographs

To ensure consistency across images, digital photographs were processed using Adobe Photoshop (Version 22.0.1, Adobe Inc., San Jose, CA, USA) to minimize variation in hairstyle, brightness and contrast, standardize vertical facial height using Na’ to Me’ soft tissue points, and adjust the background to white [20]. Three independent authors visually examined the photographs to detect any noticeable features (e.g., moles, scars) or accessories (e.g., earrings, tattoos) that might influence the evaluations. Such elements were masked during image processing. Following image adjustment, a configuration of four images per patient, consisting of pre- and post-treatment profile and frontal facial photos, was set in a landscape-oriented A4-size page as shown in Figure 1. The subsequent 36 patient photo configurations were printed and presented to the raters as described below.

### 2.3. Rater Groups

Following a previously published method [20], image sets were evaluated by four rater groups: (a) orthodontists, (b) oral and maxillofacial surgeons, (c) patients with convex profiles, and (d) laypeople. The number of rated patients per rater session was defined at 12 so that the raters would not experience fatigue or difficulty in the process [17,18,22]. Therefore, the patients under evaluation were randomly assigned into three groups of twelve (six patients from each of treatment group, with equal representation of sexes) using the website www.random.org (accessed on 23 June 2021). Each patient was then evaluated by 10 members from each rater group for the first three groups, and by 20 laypersons. For this, 30 orthodontists, 30 oral and maxillofacial surgeons, 30 Class II patients, and 60 laypersons rated the patient photos to assess perceived changes in facial appearance following the two treatment regimes.

To form the rater groups of convex profile patients and laypeople, the first white European subjects that agreed to participate were included, aiming at equal sex distribution, a wide age range between 15 and 65 years of age, and a wide range of educational level and socioeconomic status. Patients with a convex profile were recruited from the waiting area of the Postgraduate Orthodontic Clinic, with the goal of matching their age and sex to those of the post-treatment study sample (within ±3 years, or ±1 year for individuals under 19). Laypeople were selected from various locations and were not patients of the dental clinic. The first thirty specialists and final-year resident physicians who agreed to participate were included in each rater group. None of the raters had any relation to the patients. Certain raters of the specialists’ groups also participated in an analogous previous study [20].

### 2.4. Questionnaires

Each rater completed a brief personal details questionnaire before assessing the sets of photographs for each patient (Figure 1) one by one. Within each group, six cases (equally divided by sex) were randomly assigned a display format in which the initial photographs appeared on the right and the final ones on the left, while the other six cases followed the opposite configuration.

Each photograph set was paired with a validated questionnaire [17,18] containing five items. The raters were asked to assess the change in facial appearance, the change in the facial area below the nose, the change in the upper and lower lip, and the change in the chin between the left and right photos, and rate it on a 100 mm visual analogue scale (VAS) from “extremely negative” to “extremely positive” (Figure 2).

All questionnaires were administered between November 2021 and July 2022 by two researchers who were calibrated to approach the raters similarly. A pilot evaluation using a non-sample case was conducted. The raters were not informed about the specific study’s purpose or that the images depicted treated cases. Raters completed all questionnaires in a quiet, well-lit, and controlled setting to minimize distractions, and under the discreet supervision of a primary researcher (S.P.). At the rater level, the different photographic setups—each involving distinct patient sets—were assessed at separate time points, with a minimum interval of three months between evaluations to prevent potential carry-over effects.

### 2.5. Data Collection and Verification

The measurement from the starting point of the visual analogue scale (VAS) to the point marked by each rater for each question was recorded using an electronic digital caliper (Jainmed, Seoul, Republic of Korea), converting the ratings into continuous variables. These values were documented in millimeters with an accuracy of two decimal places and entered into a Microsoft Excel spreadsheet (Microsoft Corporation, Redmond, WA, USA). In instances where the final photographs were positioned on the left side, the VAS scores were adjusted by subtracting the recorded value from 100 to maintain consistency with the rest of the dataset.

The method error in measuring rater responses on the VAS was assessed previously and proved negligible [20]. Intra-rater reliability for the same questionnaire, used with a similar sample and rater population, has been tested previously and found to be satisfactory [18], and the questionnaire validity has been verified [17,18].

### 2.6. Statistical Analysis

Statistical analyses were performed with IBM SPSS statistics for Windows (Version 29.0. IBM Corp., Armonk, NY, USA), following the approach used in a previous related study [20]. Levene’s test was used to assess the homogeneity of variances, while the Shapiro–Wilk test, along with Q-Q plots and histograms, was employed to evaluate data normality. Depending on the data distribution, either parametric or non-parametric methods were applied.

Group similarity in key characteristics was assessed using the Mann–Whitney U test.

The newly generated dataset for the present study comprised 3600 questionnaire ratings collected from 150 evaluators, using a new photographic setup (simultaneous presentation of profile and frontal photos to the raters). Each patient was rated by 20 laypeople and 10 members from each of the other rater groups. Median scores per patient were calculated for each rater group and considered reliable indicators for further statistical analysis. All collected data were used, and there were no missing data or dropouts.

To assess consistency across rater groups, the intraclass correlation coefficient (ICC) was calculated using a two-way mixed-effects model with absolute agreement and average measures. ICC values exceeding 0.7 were interpreted as having strong inter-rater reliability, while values between 0.5 and 0.7 indicated moderate consistency. This assessment, together with group comparisons, supported the questionnaires’ concurrent and statistical conclusion validity.

Group differences between the orthognathic surgery and conventional orthodontic treatment cohorts were examined using a two-way multivariate analysis of variance (MANOVA). The five questionnaire responses were treated as dependent variables, whereas treatment type and rater group functioned as independent factors. When MANOVA yielded significant results, individual ANOVAs were performed for each questionnaire item, followed by post-hoc analyses using Fisher’s least significant difference (LSD) test to identify specific group differences. The present data were analyzed for the first time in this manuscript and were additionally compared with previously published data based on single-profile photo ratings of the same patient sample [20]. Differences between perceived changes in facial appearance by viewing profile only versus combined profile and frontal photos were tested with analogous multivariate analysis, followed by post-hoc tests, where applicable.

All cases were two-sided with an alpha level of 0.05. Bonferroni correction was applied for pairwise post-hoc multiple comparisons where necessary.

## 3. Results

### 3.1. Treatment Group Characteristics

The assessed patients underwent treatment between 2006 and 2020. The two treatment groups had similar Class II Division 1 malocclusions, as well as demographic and treatment duration characteristics, with a single difference in facial contour angle. This angle showed a significantly greater reduction in the orthognathic surgery group compared to the orthodontic camouflage group (Appendix A). Cephalometric analysis of additional dental and skeletal outcomes also demonstrated group comparability at baseline, with greater sagittal correction achieved through surgery compared to orthodontic camouflage (Appendix A). In group A, 13 patients received bilateral sagittal split osteotomy for mandibular advancement (one of them with additional genioplasty), 1 patient received LeFort I osteotomy, and 4 patients received bimaxillary surgery (2 of them with additional genioplasty).

### 3.2. Perceived Changes in Facial Appearance

The interrater agreement varied between good and excellent for all assessments (ICC > 0.88, Appendix A, Figure 3). The dependent variables showed equal variances across groups (Levene’s test, *p* > 0.01). Preliminary testing did not show any significant effects of the sex factor on the outcomes (*p* > 0.05), and therefore, this factor was not included in the analysis.

Multivariate analysis revealed significant differences in perceived changes in facial appearance between the two treatment groups (F = 14.63, *p* < 0.001, Pillai’s Trace = 0.36, and partial η^2^ = 0.36). No significant differences were found among rater groups (F = 1.58, *p* = 0.077, Pillai’s Trace = 0.17, and partial η^2^ = 0.06), nor were there combined effects between rater and treatment groups (F = 0.83, *p* = 0.648, Pillai’s Trace = 0.09, and partial η^2^ = 0.03). Separate ANOVAs for each dependent variable showed consistent findings with the multivariate model. The treatment group had a significant effect across all measured variables, with consistently higher VAS scores in the surgery group compared to the camouflage group. In contrast, the rater group did not significantly influence the outcomes, indicating a high level of agreement across different evaluator types. Furthermore, there were no significant interaction effects between rater and treatment groups, suggesting that all rater groups perceived the treatment differences similarly (Table 1, Figure 4). These findings support the robustness and generalizability of the observed treatment effects, regardless of rater background.

Significant positive changes in facial appearance were perceived by all rater groups following the combined orthodontic and orthognathic treatment. In contrast, no noticeable changes were observed for patients treated with orthodontics alone. The lower face, chin, and lower lip exhibited the largest mean differences in perceived changes between the surgical and camouflage treatment groups (Table 2, Figure 3 and Figure 4).

### 3.3. Comparison of Perceived Changes by Viewing Profile Only Versus Profile and Frontal Photos

In this section, previously published data based on profile assessment [20] are compared with the present data, which were generated through the simultaneous evaluation of both profile and frontal facial aspects in the same patient groups. Although statistically significant, differences among rater groups (F = 2.81, *p* < 0.001, Pillai’s Trace = 0.15, and partial η^2^ = 0.05) and between photographic setups were limited (F = 2.71, *p* < 0.021, Pillai’s Trace = 0.05, and partial η^2^ = 0.05). There were no combined effects of the rater groups and the photographic setups on the outcomes (*p* > 0.05). In contrast, there were substantial differences between the two treatment groups in the perceived changes in facial appearance (F = 31.88, *p* < 0.001, Pillai’s Trace = 0.37, and partial η^2^ = 0.37). The sex factor was not included in the analysis, since preliminary testing did not show any significant effects.

Separate ANOVAs for each dependent variable showed findings consistent with the multivariate analysis, namely no significant difference between the two photographic setups (Appendix A). The estimated marginal means for the photographic setups consistently differed on average by less than 2 VAS units and these differences were not statistically significant (Table 3). These values represent the average perceived change in facial appearance for each dependent variable, adjusting for variability among raters and treatment groups. To further explore the consistency of ratings, we performed a Spearman correlation analysis on the individual-level data across the two photographic conditions. The results show highly significant correlations for all dependent variables (*p* < 0.001), with coefficients ranging from 0.758 to 0.851, indicating strong positive associations between the two sets of ratings. These findings confirm that the inclusion of frontal views did not meaningfully alter the perceived treatment effects and support the robustness of our results.

## 4. Discussion

The present study evaluated perceived differences in facial changes induced by two distinct treatment regimens in Class II Division 1 malocclusion patients with convex facial profiles. The first approach comprised orthodontic treatment combined with orthognathic surgery and the second approach orthodontic treatment alone. The two approaches differed significantly, with the combined orthodontic and orthognathic surgery approach showing clear benefits in enhancing facial appearance despite its invasiveness and associated risks [26,27,28,29]. The lower third of the face—particularly the lower lip and chin—contributed most to the perceived differences in facial appearance between the treatment groups. However, several patients refuse to undergo orthognathic surgery due the increased costs and the fear for the operation itself, as well as for the morbidity and the complications related to the postoperative period [30,31]. On the other hand, a primary reason individuals with increased facial convexity seek treatment is to improve their facial appearance [12,13,14,30]. Therefore, these findings underscore the value of combined orthodontic and orthognathic surgery treatment for patients prioritizing aesthetic improvement, offering critical insights to guide decision-making in treatment planning. This should not be viewed as diminishing the value of orthodontic treatment alone, since it might positively affect dental and eventually smile aesthetics [32,33]. However, if the enhancement of facial appearance is a major treatment goal, which is often the case, the limitations of single orthodontic treatment should be clearly communicated to patients.

At the sample selection stage, only the initial diagnostic data of consecutively treated patients were considered, while the availability of final diagnostic data was confirmed separately. This process prioritized soft tissue parameters relevant to facial appearance, ensuring baseline similarity between treatment groups in the primary variable of interest—the soft tissue facial convexity—which is critical for valid comparisons. While there was a slight difference in severity between the groups, with the surgical cases being more severe, this reflects the clinical reality and does not undermine the validity or relevance of the comparisons. Instead, it provides a realistic basis for evaluating how these treatment approaches affect facial appearance in patients with such facial configurations. Features such as symmetry or nose shape may influence aesthetic perception, especially in frontal views. However, these characteristics are expected to be randomly distributed between the treatment groups, reflecting normal variation within the sample. As such, they are not expected to systematically bias the results in favor of one group over the other. As shown in Appendix A, both treatment groups achieved similar overjet and overbite values at the end of treatment, which fall within the range of normal occlusion. This suggests that, despite the fact that final diagnostic records were not used as inclusion criteria, both treatment modalities were capable of producing satisfactory occlusal outcomes, with no major post-treatment occlusal differences observed between the groups. Nonetheless, we acknowledge that dentoalveolar movements differed between the groups, particularly in the camouflage group, where upper incisor retraction or retroclination are commonly expected. Such movements may have influenced soft tissue changes and contributed to the observed differences in facial appearance between the groups. Treatment plans were tailored individually to each patient’s unique needs and preferences and were not part of the criteria for sample selection. This approach allowed the sample to reflect real-world clinical variability. Our analysis focused on the average morphological changes achieved by each treatment approach and their corresponding perceptions. As a result, the applicability of the findings is grounded in the actual morphological outcomes rather than in the specifics of individual treatment plans and responses, which may vary even among patients with similar clinical conditions [34].

The effectiveness of orthodontic treatment alone in meeting this need has been questioned, even among growing patients aiming at enhancing mandibular growth [15,16,17,18]. Previous research on growing patients treated with functional appliances reported only modest improvements in profile appearance. Perceptions of different groups were consistent [17] and the small improvements attributed to treatment diminished when profile and frontal facial images were presented simultaneously to the raters [18]. The aforementioned studies reported a modest improvement of approximately 10% in the facial appearance of all tested groups, attributed primarily to the maturation from preadolescence to adolescence. On the contrary, significant improvements of about 20% in facial profile appearance were consistently perceived by different groups of evaluators in non-growing patients that were subjected to orthognathic surgery [20] compared to no improvement with orthodontic camouflage treatment. The present study highlighted that the considerable differences between treatment groups were similarly perceived when presenting simultaneously facial and profile photos to the raters. This is a noteworthy finding, especially given that the orthognathic intervention only modifies facial morphology, which is just one among several factors that could influence the perception of facial appearance [35]. Previous studies have shown that assessments are modified when different facial views are presented to the raters [18,36,37]. The fact that the considerable improvement perceived in facial profiles remained similarly perceivable when frontal photos were also presented indicates that the changes in overall facial appearance were fundamental. Therefore, the present study offers important insights to the actual treatment effects on facial appearance so that the patients can receive evidence-based information regarding the expected outcomes, and the anticipated positive impact of treatment on social, psychological, and even economic outcomes. This will facilitate evidence-based decision making during individualized treatment planning relative to the important outcome of facial appearance, which should be considered along with a number of other factors [38].

Previous research has shown that raters with diverse backgrounds perceive certain facial outcomes differently [21,34,39] and that although objectively measured phenotypic traits contribute significantly to facial attractiveness, a series of other factors is also important [35]. Importantly, the absence of significant effects from the rater group and of any interaction between rater and treatment group indicates that perceptions of facial changes were consistent across evaluator types. This suggests that the observed improvements in facial appearance following orthognathic surgery were perceived similarly by professionals and laypersons alike. Such consistency enhances the external validity of the findings, as they are unlikely to be biased by rater background. The significant effects were solely attributed to the treatment type, with surgical intervention consistently yielding higher ratings of perceived improvement. The use of actual patient images instead of largely modified ones is considered a closer approximation of the reality of human interactions, and thus, of associated effects [40,41]. Actual patient photos have been rarely used previously for such outcomes, and the existing studies present conflicting findings [42]. The study of Shell and Woods [43] showed similar effects of both treatments on facial attractiveness, whereas the study of Proffit et al. [44] identified only minor differences of about 5%, favoring surgical outcomes. Both studies rated sets of frontal and profile facial images for facial attractiveness, but assessed separately the pre- and post-treatment conditions. Rater groups were also not precisely defined and analyzed. We used slightly modified facial photos to retain the original appearance of individuals, while limiting the effects of confounding factors such as hairstyle and prominent marks or jewelry. Moreover, we applied a robust methodology regarding questionnaire validity and different rater groups that are important from various perspectives in decision-making or treatment impact [17,18,20,22]. Another strength of the present study is the simultaneous presentation of the pre- and post-treatment images asking the raters to assess changes after randomizing the treatment phase status. With this design, various individual factors that could potentially confound the assessments—such as skin color, texture, hair color, hairstyle, and certain local morphological features [1,35]—are controlled, enhancing the precision of the outcome, which specifically focuses on the perceived impact of morphological changes on facial appearance. Although the surgery cases appeared to be slightly more severe at baseline, the differences between the groups were relatively small and not statistically significant, ensuring their baseline comparability for analysis, particularly regarding soft tissue parameters relevant to facial appearance. We acknowledge that there was a slight, statistically significant difference in skeletal severity between the groups, which might reflect clinical reality and should be taken into account during outcome interpretation. The present study identified clear, substantial differences that remained consistent across different types of raters and facial views [20]. These differences likely emerged as a result of the methodological considerations employed.

The absence of an a priori power calculation could be considered a study limitation, particularly for testing interaction terms. However, a post-hoc analysis confirmed that the sample size was sufficient to achieve adequate statistical power for detecting a medium effect size. We determined a reasonable sample size based on empirical evidence and resource availability, balancing feasibility in terms of both patient and rater numbers. All data are comprehensively presented, allowing readers to critically evaluate the outcomes. Consistent with previous similar studies [17,18,22], the detection of significant differences between treatment groups suggests sufficient power to address the primary outcomes. We did not conduct a specific study to determine the optimal number of raters, but based our selection on the assumption that the median response from the chosen number of raters would provide a representative assessment for each patient. While no universally established number of raters exists for facial appearance assessment, our decision was informed by empirical evidence. Saito et al. explored the effective number of subjects and raters for inter-rater reliability studies, supporting that our chosen sample size does not introduce significant sampling issues [45]. Based on this and our findings, we believe our approach strikes a reasonable balance between reliability and feasibility. The sample comprised individuals of white European ancestry, as they represented the vast majority of patients treated at the sample collection site. Including a small number of individuals from other racial backgrounds would not have allowed for proper control of potential confounding effects due to this factor. The present results need to be tested in diverse racial groups to determine their generalizability and to explore potential variations in outcomes that may be influenced by genetic, cultural, or environmental factors. The inclusion criteria were defined to exclude extremes in facial morphology, where we would expect greater effects by the surgical approach, but this, as well as the respective effects of camouflage orthodontic treatment, remains to be tested. This study did not test the factors underlying the decisions made by the patients and their doctors, but focused on the perceived morphological outcomes of the interventions. Pre-treatment facial attractiveness and facial morphology were not thoroughly assessed and should be investigated in future studies as potential mediators of these findings. Additionally, a separate evaluation of frontal views was not performed. In this project, we prioritized two image configurations: profile-only and combined frontal with profile views. This decision was based on the clinical relevance of the profile in treatment evaluation and the need to reflect real-life facial perception. Adding a third, frontal-only assessment would have required additional rater sessions spaced sufficiently to avoid confounding effects. Future studies could benefit from independently assessing frontal views to explore their specific contribution to perceived outcomes. The present assessment used static images. Functional assessment through actual interactions or presentations of videos might have modified the outcomes. Finally, this study assessed exclusively changes in facial appearance, not accounting for dental or smile aesthetics, which might have affected the outcomes [33].

Future research could incorporate three-dimensional video recordings or live assessments that include dynamic evaluations of functioning during interpersonal communication. Additionally, studies could investigate the long-term impact of combined treatment on patient satisfaction and functional outcomes, offering a more comprehensive understanding of its overall benefits.

## 5. Conclusions

This study found that combined orthodontic treatment and orthognathic surgery for managing facial convexity resulted in noticeable enhancements in facial appearance, as perceived by various rater groups. In contrast, orthodontic treatment alone did not lead to significant perceived changes. Different rater groups, including laypeople, evaluated facial changes in a similar manner.

These findings have significant clinical implications for adult patients with Class II Division 1 malocclusion and facial convexity and should be carefully considered during patient consultations and individualized treatment planning. For patients seeking significant facial appearance improvement, clinicians must clearly communicate the limitations of orthodontic treatment alone and emphasize the potential benefits of combining it with orthognathic surgery. Transparent discussions regarding anticipated outcomes are crucial for managing patient expectations. Orthognathic surgery may be a suitable option for individuals primarily seeking facial appearance enhancement, while those with fewer aesthetic concerns may consider orthodontic treatment alone. In all cases, a multidisciplinary approach and comprehensive pre-treatment evaluation are essential to optimize patient satisfaction.

## Figures and Tables

**Figure 1 jcm-14-04277-f001:**
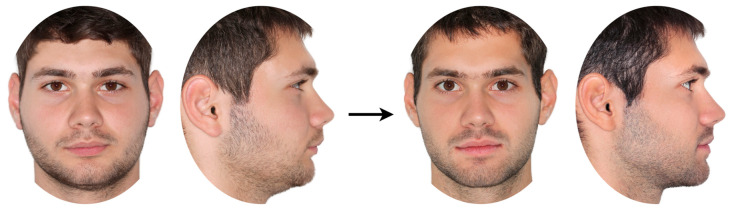
Example of patient images shown for rating. In this case, pre-treatment images are on the left and post-treatment images on the right. For half of the cases, this orientation was reversed (post-treatment on the left, pre-treatment on the right), following a stratified randomization protocol implemented via https://www.random.org/.

**Figure 2 jcm-14-04277-f002:**
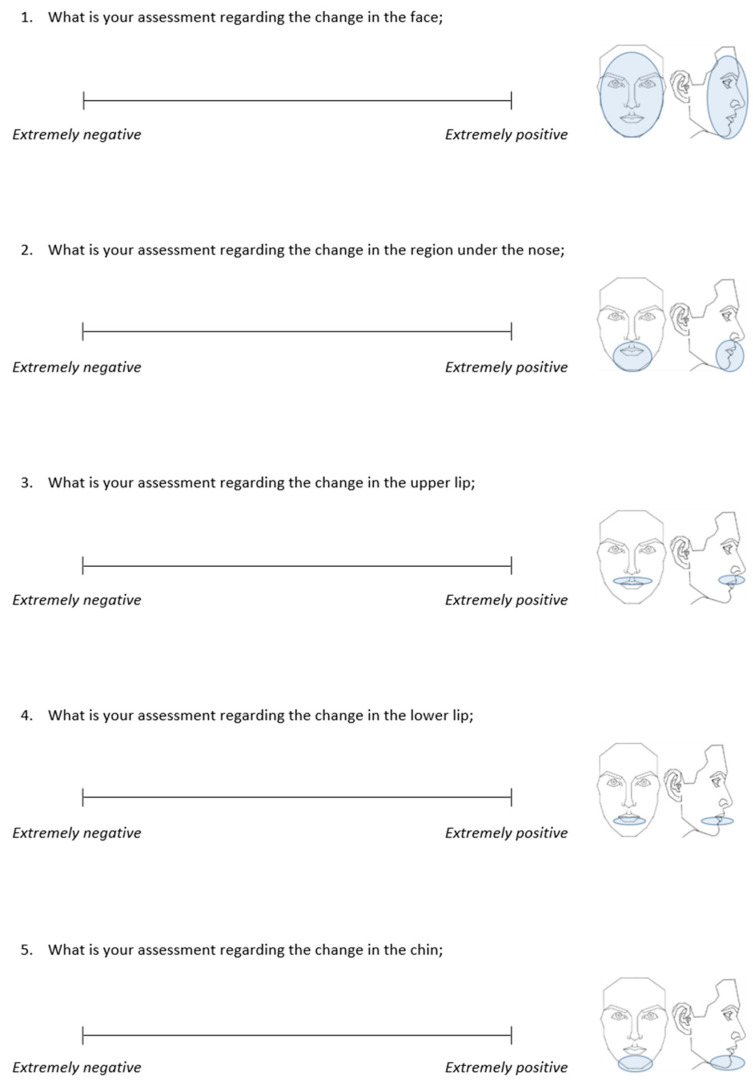
The administered questionnaire to assess changes in facial appearance from pre- to post-treatment through the visual analogue scale (VAS).

**Figure 3 jcm-14-04277-f003:**
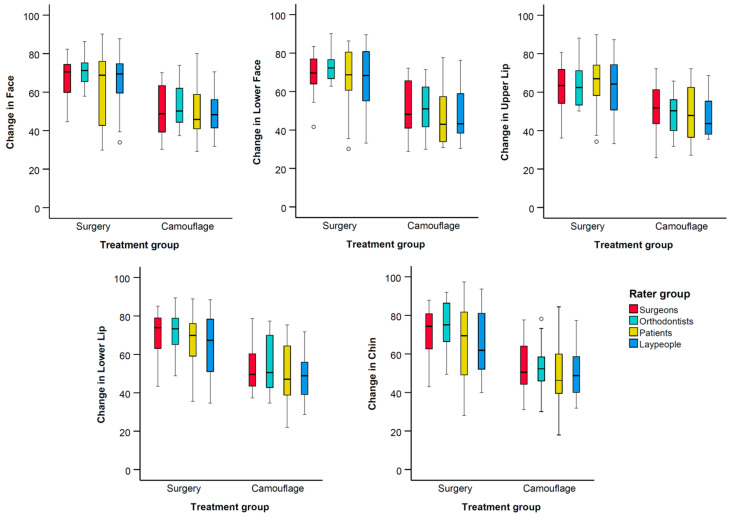
Box plots showing the perceived changes in facial appearance for each treatment group, evaluated by the different raters. The plots are clustered by treatment group and display VAS values on the y-axis. The upper end of the black line indicates the maximum value, while the lower end represents the minimum value. The box displays the interquartile range, and the horizontal black line within the box marks the median value. Outliers exceeding ±3 standard deviations are depicted as black circles.

**Figure 4 jcm-14-04277-f004:**
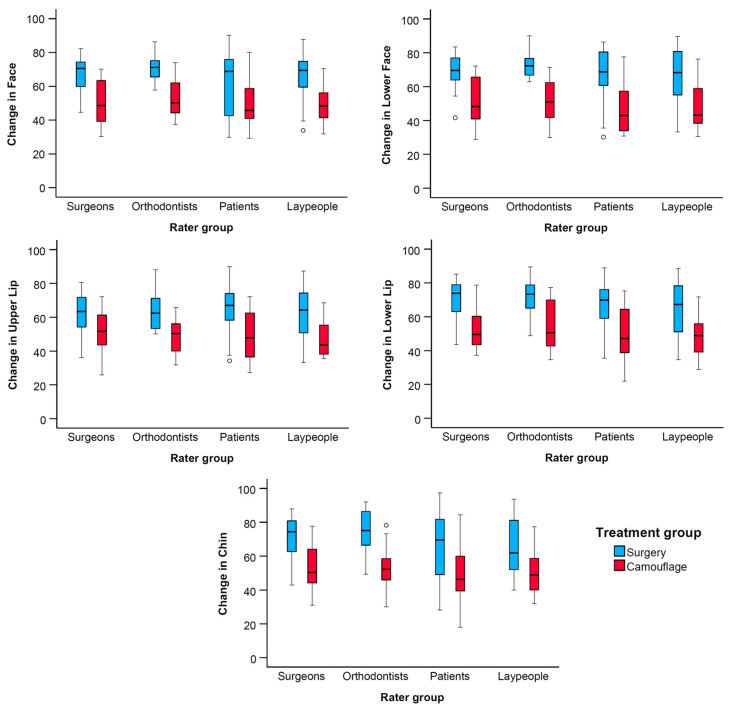
Box plots showing the perceived changes in facial appearance for each treatment group, evaluated by the different raters. The plots are clustered by rater group and display VAS values on the y-axis. The upper end of the black line indicates the maximum value, while the lower end represents the minimum value. The box displays the interquartile range, and the horizontal black line within the box marks the median value. Outliers exceeding ±3 standard deviations are depicted as black circles.

**Table 1 jcm-14-04277-t001:** Results of the ANOVAs testing the effects of the rater group and treatment group, as well as their interactions, on the perceived facial changes (dependent variables).

Source	Dependent Variable	df	F	Sig.
Rater group	Face ^a^	3	1.17	0.322
	Lower face ^b^	3	1.61	0.190
	Upper lip ^c^	3	0.17	0.916
	Lower lip ^d^	3	1.84	0.143
	Chin ^e^	3	1.87	0.138
Treatment group	Face	1	58.53	<0.001
	Lower face	1	73.78	<0.001
	Upper lip	1	48.49	<0.001
	Lower lip	1	53.76	<0.001
	Chin	1	49.17	<0.001
Rater group × treatment group	Face	3	0.18	0.906
	Lower face	3	0.19	0.900
	Upper lip	3	0.21	0.891
	Lower lip	3	0.01	0.999
	Chin	3	0.34	0.799

^a^ R squared = 0.31 (adjusted R squared = 0.28), ^b^ R squared = 0.37 (adjusted R squared = 0.33), ^c^ R squared = 0.27 (adjusted R squared = 0.23), ^d^ R squared = 0.30 (adjusted R squared = 0.27), and ^e^ R Squared = 0.29 (adjusted R squared = 0.25). df: degrees of freedom. F: F-value. Sig.: significance shown as *p*-values.

**Table 2 jcm-14-04277-t002:** Estimated marginal means per dependent variable for each treatment group and statistical comparisons.

Dependent Variable	Treatment Group	Mean	Standard Error	95% Confidence Interval		Mean Difference (Surgery–Camouflage)	Standard Error	Sig.
				Lower Bound	Upper Bound			
Face	Camouflage	50.81	1.54	47.77	53.85	16.62	2.17	<0.001
	Surgery	67.44	1.54	64.40	70.47			
Lower face	Camouflage	49.61	1.59	46.46	52.75	19.30	2.25	<0.001
	Surgery	68.91	1.59	65.77	72.05			
Upper lip	Camouflage	49.00	1.51	46.02	51.98	14.84	2.13	<0.001
	Surgery	63.85	1.51	60.87	66.83			
Lower lip	Camouflage	51.37	1.66	48.09	54.64	17.19	2.34	<0.001
	Surgery	68.56	1.66	65.28	71.84			
Chin	Camouflage	51.61	1.74	48.18	55.05	17.22	2.46	<0.001
	Surgery	68.83	1.74	65.40	72.27			

Sig.: Significance shown as *p*-values.

**Table 3 jcm-14-04277-t003:** Estimated marginal means per dependent variable for each photographic setup of the facial images and statistical comparisons.

Dependent Variable	Photographic Setup	Mean	Standard Error	95% Confidence Interval		Mean Difference (Profile–Frontal and Profile)	Standard Error	Sig.
				Lower Bound	Upper Bound			
Face	Frontal and profile	59.12	1.00	57.16	61.09	1.19	1.41	0.401
	Profile	60.31	1.00	58.35	62.28			
Lower face	Frontal and profile	59.26	1.05	57.20	61.32	1.88	1.48	0.206
	Profile	61.13	1.05	59.07	63.20			
Upper lip	Frontal and profile	56.43	1.02	54.42	58.43	−1.04	1.44	0.471
	Profile	55.38	1.02	53.38	57.39			
Lower lip	Frontal and profile	59.96	1.12	57.75	62.17	0.65	1.59	0.684
	Profile	60.61	1.12	58.40	62.82			
Chin	Frontal and profile	60.22	1.11	58.03	62.42	1.61	1.58	0.308
	Profile	61.83	1.11	59.64	64.02			

Sig.: Significance shown as *p*-values.

## Data Availability

The datasets generated and/or analyzed during the current study are available from the corresponding author on reasonable request.

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
