# Peer review of "Aesthetic Impact of Orthognathic Surgery vs. Orthodontic Camouflage in Class II Division 1 Patients with Convex Facial Profile: A Follow-Up Using Combined Frontal and Profile Views"

_jcm, 2025, doi:10.3390/jcm14124277_

Round 1
Reviewer 1 Report
Comments and Suggestions for Authors
It is also important to say also in the TITLE, results, and conclusion that it is a class II/1 you are considering.
It is not clear what you mean by: "Half of the cases were displayed in the opposite
orientation, with the post-treatment status on the left and the pre-treatment status on the right".
Figure 2 can be moved in the supplementary file.
The effects of rater group and treatment group, as well as their interactions, on the perceived facial changes should be explained better.
Please explain the estimated marginal means per dependent variable for each photographic setup of the facial images. Are there some correlations?
Discussion: The present study evaluated perceived differences in facial changes induced by two distinct treatment regimens in patients with convex profiles - IN class 2/1!!!
What is the originality?
Please describe limitations.
The conclusion should be more specific rather than: These findings have significant clinical implications for adult patients with facial convexity and should be carefully considered during patient consultations and individualized treatment planning. HOW?
Reviewer 2 Report
Comments and Suggestions for Authors
Dear Editor, thank you for the opportunity to review this scientific paper. The purpose of this study is to assess the treatment outcomes of combined orthognathic/orthodontic treatment versus orthodontic camouflage treatment in terms of the perceived change in facial profile appearance. The topic is interesting, but there are some important inaccuracies:
- Highlight which facial subregions (e.g., chin, lower lip) showed the largest perceived changes and contributed most to the overall aesthetic improvement.
- Discuss more thoroughly the impact of including frontal views, especially considering that results did not significantly differ from the profile-only setting.
- Address possible confounding visual elements present in frontal photos (e.g., smile, symmetry, nose shape) that may affect aesthetic perception.
- Please follow the STROBE Checklist and attach as supplementary
- Pre- and post-treatment photos were randomly positioned (left/right), but no analysis was conducted to assess if this influenced perception. Please add or justify this point.
- There is no clinical experience split (e.g., residents vs. final year residents), which could influence the aesthetic judgment. Please add or justify this point.
- The paper focuses exclusively on facial aesthetics, but does not discuss functional orthodontic aspects (e.g., improved occlusion, post-treatment stability, masticatory function). It would be useful to add or at least comment on the differences in dentoalveolar movements in the two approaches and the possible side effects of camouflage, such as excessive retraction of the upper incisors and worsening of the labial profile.
- Why was a VAS scale used, and not a 5-point Likert scale?
- Considering that this study is an implementation of the previous one, it would be useful to ask the evaluators to evaluate the frontal profile separately, and the frontal and lateral profile together so as to have a better perception of the impact on the evaluation of the frontal profile as well. Please justify this point.
Round 2
Reviewer 1 Report
Comments and Suggestions for Authors
The paper has been improved.
Group differences between the orthognathic surgery and conventional orthodontic treatment cohorts have been better described.
The change in facial appearance for each dependent variable has been clearly presented.
The discussion has been expanded and improved.
The conclusion is clear.
Author Response
Thank you very much for your positive evaluation and for acknowledging the improvements made to the manuscript. We are glad that the revised version addresses the key aspects you highlighted.
Reviewer 2 Report
Comments and Suggestions for Authors
Dear Editor,
Thank you for the opportunity to review the revised version of this manuscript and the authors’ responses to the reviewers' comments. The authors have addressed many of the points raised with clarity and rigor. However, I would like to submit the following follow-up comments on two specific points, which in my opinion still warrant further consideration:
Follow-up on Comment 4 (STROBE Checklist):
I thank the authors for their response. However, I would like to respectfully point out that including the full STROBE checklist as a Supplementary file is a standard requirement in many journals, precisely to ensure transparency and allow readers to easily verify the completeness of reporting. The checklist does not unnecessarily expand the manuscript and its inclusion would strengthen the methodological rigor of the paper. I would encourage the authors to reconsider and to add the full checklist as Supplementary material, in line with best practices for reporting observational studies.
Follow-up on Comment 9 (Separate evaluation of frontal view):
I appreciate the authors' explanation regarding their choice to present combined frontal and profile images. However, my original suggestion was aimed at improving the methodological depth of the study by also including a separate evaluation of the frontal view, which could have been implemented in parallel, without necessarily introducing confounding factors. The current response does not fully justify why this option was not adopted, despite its potential to provide additional insights into the perceptual impact of the frontal profile. I would therefore recommend that the authors either provide a clearer methodological justification or explicitly acknowledge this as a limitation in the Discussion section.
In conclusion, I believe that with these additional clarifications and adjustments, the manuscript would reach a higher level of methodological transparency and scientific rigor.
Thank you once again for the opportunity to contribute to the review process.
Kind regards,
Author Response
Dear Editor, Dear Reviewers,
Thank you once again for reviewing our study and for the constructive feedback that helped improve the quality of our work. We hereby resubmit our revised manuscript along with our point-by-point responses. We hope the current version adequately addresses the remaining concerns raised by Reviewer 2.
Reviewer 2
Comment 1: The authors have addressed many of the points raised with clarity and rigor. However, I would like to submit the following follow-up comments on two specific points, which in my opinion still warrant further consideration:
Follow-up on Comment 4 (STROBE Checklist):
I thank the authors for their response. However, I would like to respectfully point out that including the full STROBE checklist as a Supplementary file is a standard requirement in many journals, precisely to ensure transparency and allow readers to easily verify the completeness of reporting. The checklist does not unnecessarily expand the manuscript and its inclusion would strengthen the methodological rigor of the paper. I would encourage the authors to reconsider and to add the full checklist as Supplementary material, in line with best practices for reporting observational studies.
Response 1: We are happy to see that the reviewer was satisfied with the revision and did our best to address adequately the remaining concerns.
The STROBE checklist has been completed and submitted according to the reviewer’s suggestion (Supplementary File S1).
Comment 2: Follow-up on Comment 9 (Separate evaluation of frontal view):
I appreciate the authors' explanation regarding their choice to present combined frontal and profile images. However, my original suggestion was aimed at improving the methodological depth of the study by also including a separate evaluation of the frontal view, which could have been implemented in parallel, without necessarily introducing confounding factors. The current response does not fully justify why this option was not adopted, despite its potential to provide additional insights into the perceptual impact of the frontal profile. I would therefore recommend that the authors either provide a clearer methodological justification or explicitly acknowledge this as a limitation in the Discussion section.
In conclusion, I believe that with these additional clarifications and adjustments, the manuscript would reach a higher level of methodological transparency and scientific rigor.
Response 2: We thank the reviewer for this thoughtful follow-up. We agree that a separate evaluation of frontal views could indeed enhance the methodological depth of the study while effectively controlling potential confounding factors. However, our primary focus was on profile image configurations, as the profile view is most directly related to the treatments under investigation. To approximate real-life facial perception and clinical evaluation more closely, we included a combined presentation of frontal and profile views.
To minimize potential carry-over effects, different photographic setups—each comprising distinct patient sets—were assessed at separate time points, with a minimum three-month interval between evaluations. Introducing a third condition (frontal view only) would have required an additional set of evaluations, spaced adequately to prevent bias, and this would have significantly increased the demand on rater time and resources.
For these practical reasons, we limited the study to two configurations: a) profile-only and b) combined frontal + profile views. We now explicitly acknowledge this as a limitation in the revised Discussion section (second-to-last paragraph, marked in red) and agree that future research could usefully explore the isolated perceptual impact of frontal views.
